# A Novel Generalized Nested Array MIMO Radar for DOA Estimation with Increased Degrees of Freedom and Low Mutual Coupling

**DOI:** 10.3390/s24123952

**Published:** 2024-06-18

**Authors:** Zhongtian Yang, Zhengyang Bi, Ye Chen, Honghao Hao

**Affiliations:** 1School of Energy and Power Engineering, Beihang University, Beijing 102200, China; buaayztian@buaa.edu.cn; 2College of Advanced Technology, Xi’an Jiaotong-Liverpool University, Suzhou 215000, China; zhengyang.bi22@student.xjtlu.edu.cn; 3College of Electronic Information Engineering, Nanjing University of Aeronautics and Astronautics, Nanjing 210016, China; haowenjie1025@163.com

**Keywords:** multiple-input multiple-output (MIMO) radar, nested array, direction of arrival estimation, degree of freedom, mutual coupling

## Abstract

In array signal processing, the mutual coupling among physical sensors can inevitably affect the estimation of the direction of arrival (DOA). Despite the fact that multiple-input and multiple-output (MIMO) radar can provide greater degrees of freedom (DOFs), the influence of mutual coupling is largely overlooked in many current MIMO radar designs. To tackle this issue, we propose the utilization of a generalized nested array (GNA) in transmitter array and we introduce an expansion factor into the nested array in the receiver array. Thereby, a novel GNA-MIMO radar is put forward. The proposed MIMO radar offers $O(N^{4})$ consecutive DOFs with *N* sensors and avoids the adverse effects of high mutual coupling caused by closely located sensors. Furthermore, we derive the closed-form expressions for the position of physical sensors and the attainable consecutive DOFs of the proposed MIMO radar. Through simulation experiments, we demonstrate the superior accuracy of the proposed MIMO configuration in DOA estimation and angle resolution under the condition of mutual coupling effect.

## 1. Introduction

In the field of array signal processing, direction of arrival (DOA) estimation is a crucial topic with significant importance in wireless communication, radar and medical imaging, etc., [1,2,3,4]. Multi-input multi-output (MIMO) radar is an innovative radar system that was initially proposed in 2003 [5]. In the past decades, MIMO radar has drawn substantial concern and is becoming an increasingly hot topic [3,6]. One of the advantages of MIMO radar is the ability to create a virtual array with large aperture and increased degrees of freedom (DOFs) by utilizing multiple transmitter and receiver antennas [7]. The number of DOFs is one of the key parameters to measure the performance of DOA in MIMO radar systems, which represents the ability of radar to distinguish and process multiple targets simultaneously. DOFs are mainly determined by the number of antennas, the distance between antennas and the arrangement of sensors. By increasing DOFs, the resolution, detection accuracy and anti-interference ability of the radar system can be significantly improved. However, the current research on MIMO radar systems mainly focuses on ways to achieve higher DOFs while few of them study the mutual coupling effect between physical sensors. The impact of mutual coupling cannot be neglected in practice and can degrade the performance of DOA estimation, especially when physical sensors are closely located [8].

Various design methods have been put forward to design MIMO radar array configurations with better properties. Initially, the design of minimal redundancy (MR) MIMO radar was proposed to reduce sensor redundancy [9,10,11,12]. Nonetheless, this method suffers from high computational complexity, leading to increased costs. Subsequently, the advancement of nested array (NA) [13], coprime array (CA) [14] and other sparse linear arrays [15,16,17] have offered a new perspective in MIMO radar system design. For example, Qin et al. proposed a design method for nested MIMO radar [18] and co-prime MIMO radar [19] through the utilization of two subarrays of NA (or CA) in the transmitter and receiver arrays, respectively. However, the above-mentioned MIMO radar has the limitation of restricted DOFs, the maximal DOFs of which can only reach $O(N^{2})$ with *N* physical sensors. To increase DOFs, the concept of difference co-array of sum co-array (DCSC) [12] as well as the generalized sum co-array of difference co-array (GSDC) [13] were proposed. These two methods gained significant attention for the ability to achieve $O(N^{4})$ DOFs with only *N* physical sensors. Zheng et al. proposed a nested MIMO radar with a sparsely extended array geometry, the transmitter array of which is composed of extended nested arrays by introducing an expansion factor [20]. Likewise, a generalized co-prime MIMO radar was proposed with generalized co-prime arrays by introducing an expansion factor to the transmitter array [21]. Later in [22], a flexible MIMO radar was proposed to enhance the element spacing of transmitter and receiver arrays by generalizing the coprime expansion factors. It is worth mentioning that the above conventional nested and co-prime MIMO radars represent special cases of flexible MIMO radar. Additionally, Zhang et al. proposed a novel generalized nested MIMO radar by replacing the nested array with an extended nested array to achieve more DOFs [23]. Furthermore, to gain more attainable DOFs, Shaikh et al. devised a generalized multi-level MIMO radar configuration [24].

However, the above-mentioned MIMO radar systems neglected the influence of mutual coupling. Consequently, despite the great number of DOFs, these systems suffer from significant degradation in DOA performance caused by severe mutual coupling in practice. In [25,26], a UCA-MIMO radar was proposed by using unfolded co-prime arrays (UCA) in the transmitter and receiver arrays. Although this design only achieves $O(N^{2})$ DOFs, the advantage lies in the sparsely located sensors. In [27], by using uniform linear arrays in the transmit-receiver array and introducing coprime factors, a generalized two-level nested array (GTNA) MIMO radar was proposed. Benefiting from the coprime factor, the element spacing continuously increases while the consecutive DOFs gradually decrease, thus reaching a balance between a great number of consecutive DOFs and a weak mutual coupling effect. However, the optimal selection of coprime factors was not discussed. In [28], a UGCA-MIMO radar was proposed by utilizing an unfolded generalized co-prime array (UGCA) in the transmitter and receiver arrays. However, no closed-form expression was provided. To address these issues and limitations, this paper puts forward the design of a GNA-MIMO radar. The proposed structure utilizes a GNA in the transmitter array and a prototype NA with a sparse expansion factor in the receiver array. The main contributions of this paper are as follows:

(1) The proposed GNA-MIMO radar reaches the balance between both a great number of consecutive DOFs and a weak mutual coupling leakage effect. Specifically, it achieves $O(N^{4})$ consecutive DOFs with only *N* physical sensors with no need for closely located physical sensors in the transmitter or receiver arrays. This achievement is noteworthy as it surpasses the capabilities of most previously devised MIMO radars.

(2) We derive the closed-form expressions for the location of physical sensors and the attainable consecutive DOFs for the proposed GNA-MIMO radar. Meanwhile, we present the design principle for GNA-MIMO radar in Section 3.3. In contrast, the UGCA-MIMO radar designed in [29] lacks closed-form expressions and is not suitable for all possible numbers of sensors. Simulation results also prove that the proposed GNA-MIMO radar has a weaker coupling effect.

The paper is outlined as follows: Section 2 introduces the data model of the monostatic MIMO radar. The GNA-MIMO radar geometry is proposed in Section 3. In Section 4, sufficient theoretical analysis is provided. Section 5 provides lots of numerical results and Section 6 concludes.

Throughout this paper, the upper-case bold and lower-case bold characters, respectively, denote matrices and vectors; ${\left(\cdot \right)}^{T}$, ${\left(\cdot \right)}^{*}$ and ${\left(\cdot \right)}^{H}$ denote transpose, conjugate and conjugate transpose; ⊙ and ⊗ represent Khatri–Rao product and Kronecker product; $I_{N}$ denotes an N×N identity matrix; E[·] denotes the expectation operator; diag(·) denotes the diagonal operation; vec(·) stands for the vectorization operator.

## 2. Signal Model

A narrow-banded monostatic MIMO radar system is composed of an *M*-element transmitter array and an *N*-element receiver array. The transmitter array emits multiple orthogonal signals while the receiver array captures the reflected signals.

Assuming that there are *K* far-field incoherent targets, where $\theta_{k}$ denotes the direction of the *k*-th target. Then the MIMO steering vector $a(\theta_{k})$ equals to [20]
(1)$$a\left(\theta_{k}\right)=a_{r}\left(\theta_{k}\right)⊗a_{t}\left(\theta_{k}\right),$$
where $a_{r}\left(\theta_{k}\right)$ and $a_{t}\left(\theta_{k}\right)$ are the steering vectors of the receiver and transmitter corresponding to the *k*-th target, respectively. And $a_{r}\left(\theta_{k}\right)$ and $a_{t}\left(\theta_{k}\right)$ can be given as [20]
(2)$$\;\;a_{r}\left(\theta_{k}\right)\;=\;{\left[e^{\frac{j2\pi d_{1}sin\theta_{k}}{\lambda }},\;\cdots \;,e^{\frac{j2\pi d_{N}sin\theta_{k}}{\lambda }}\right]}^{T},$$
(3)$$\;a_{t}\left(\theta_{k}\right)\;=\;{\left[e^{\frac{j2\pi d_{1}sin\theta_{k}}{\lambda }},\;\cdots \;,e^{\frac{j2\pi d_{M}sin\theta_{k}}{\lambda }}\right]}^{T},$$
where λ denotes the wavelength, $d_{n}\left(n=1,2,\cdots ,N\right)$ and $d_{m}^{\prime }$(m=1,2,⋯,M) stands for the sensor locations in the receiver and transmitter arrays.

Further, $a(\theta_{k})$ can be written as [21]
(4)$$\begin{array}{c} a\left(\theta_{k}\right)=[e^{j2\pi {(d}_{1}+d_{1}^{\prime })sin\theta_{k}/\lambda },e^{j2\pi {(d}_{1}+d_{2}^{\prime })sin\theta_{k}/\lambda }\\ ,\cdots ,e^{j2\pi {(d}_{2}+d_{1}^{\prime })sin\theta_{k}/\lambda },e^{j2\pi {(d}_{2}+d_{2}^{\prime })sin\theta_{k}/\lambda }\\ ,\cdots ,e^{j2\pi {(d}_{N}+d_{M}^{\prime })sin\theta_{k}/\lambda }], \end{array}$$
where the MIMO steering vector $a(\theta_{k})$ is a MN×1 steering vector.

Subsequently, taking mutual coupling into account, the signal model of the received signal can be formulated as [28]
(5)$$\begin{array}{cc} X(t) & =C_{t}⊗C_{r}\sum_{k=1}^{K}a_{t}\left(\theta_{k}\right)⊗a_{r}\left(\theta_{k}\right)s\left(t\right)+n\left(t\right)\\ & =\mathrm{CA}s(t)+n(t), \end{array}$$
where $s\left(t\right)=[s_{1}\left(t\right),s_{2}\left(t\right),\cdots ,s_{K}\left(t\right)]$ represents the signal source vector, n(t) represents the zero-mean Gaussian white noise vector, $A=[a\left(\theta_{1}\right),\cdots ,a\left(\theta_{K}\right)]$ represents the steering matrix and $a\left(\theta_{k}\right)=a_{t}\left(\theta_{k}\right)⊗a_{r}\left(\theta_{k}\right)$ is the steering vector of the *k*-th signal. The mutual coupling matrix $C=C_{t}⊗C_{r}$. The mutual coupling matrice C can be approximated as [15]
(6)$$C_{ij}=\left\{\begin{array}{c} c_{|p_{i}-p_{j}|},\left|p_{i}d-p_{j}d\right|\leq B\\ 0,\;\;\;\;\;|p_{i}d-p_{j}d|>B \end{array}\right.$$
where $C_{ij}$ denotes the element located in the *i*-th row and *j*-th column of the matrix C and $p_{i}d$ denotes the position of *i*-th sensor (*i* = 1, 2, ...*N*). $1=c_{0}>|c_{1}\left|>\cdots >\right|c_{B}\left|>\right|c_{B+1}|=0$, $c_{l}=c_{1}e^{-j(l-1)\pi /8}/l$ for 2≤B≤100 [30]. Subsequently, the coupling leakage can be measured by the parameter [30]
(7)$$L\left(M\right)=\frac{||C-{I||}_{F}}{{\left||C|\right|}_{F}}$$
where I represents the identity matrix with the same order as the matrix C.

The covariance matrix of the received signal can be calculated as [20]
(8)$$R_{x}=E\left[X\left(t\right)X^{H}\left(t\right)\right]=AR_{s}A^{H}+\sigma_{n}^{2}I_{N^{2}},$$
where $R_{s}=E\left[s\left(t\right)s^{H}\left(t\right)\right]=diag\left(\left[\sigma_{1}^{2},\ldots ,\sigma_{K}^{2}\right]\right)$ and $\sigma_{k}^{2}$ denotes the power of the *k*-th source. In practice, the covariance matrix is calculated through a certain number of snapshots, ${\hat{R}}_{x}=\left(1/L\right)\sum_{t=1}^{L}X\left(t\right)X^{H}\left(t\right)$, where *L* is the number of snapshots.

Vectorizing the covariance matrix in (Equation 8) yields [20]
(9)$$z=vec\left(R_{x}\right)=vec\left(A_{0}p\right)+\sigma_{n}^{2}vec\left(I_{N^{2}}\right),$$
where $A_{0}=A^{*}⊙A=\left[a^{*}\left(\theta_{1}\right)⊗a\left(\theta_{1}\right),a^{*}\left(\theta_{2}\right)⊗a\left(\theta_{2}\right),\cdots ,a^{*}\left(\theta_{K}\right)⊗a\left(\theta_{K}\right)\right]$ is a new steering matrix including a lot of virtual array elements; $p={\left[\sigma_{1}^{2},\ldots ,\sigma_{K}^{2}\right]}^{T}$. From (Equation 9), the vector z indicates a new received signal corresponding with the new steering matrix $A_{0}$ with a single snapshot.

## 3. Generalized Nested Array MIMO Radar Configuration

In this section, we present a concise review of sum co-array and difference co-array. Second, the reformulation of GNA is shown. Later, the design process of the Generalized Nested Array MIMO radar is presented. At last, the closed-form expression of consecutive virtual array elements and the available consecutive DOFs are derived in detail.

### 3.1. Review of Sum Co-Array and Difference Co-Array

**Definition** **1.**
*The position set of sensors for a given M−element array is denoted by set S:*

$$S=\{p_{1},p_{2},\cdots ,p_{M}\}d.$$

*where M represents the number of elements and d denotes the unit element spacing, d=λ/2. In subsequent representations in this paper, we omit the parameter d.*


**Definition** **2.**
*The sum co-array generated by the arrays $S_{a}$ and $S_{b}$ can be defined by the unique elements in the set $S_{sum}$, which can be written as [18]*

$$S_{sum}=\left\{u+u^{\prime }|u\in S_{a},u^{\prime }\in S_{b}\right\}.$$



**Definition** **3.**
*The difference co-array generated by the array $S_{u}$ is symmetric and can be similarly defined by the unique elements in the set $S_{diff}$, which can be given as [31]*

$$S_{diff}=\left\{u-u^{\prime }|u,u^{\prime }\in S_{u}\right\}.$$



**Definition** **4.**
*Given an array S, the number of elements in the consecutive segment of its difference co-array $S_{diff}$ is defined as “consecutive DOFs”.*


For instance, the ULAs $S_{a}=\left\{0,1,2\right\}$ and $S_{b}=\left\{0,3\right\}$ can obtain the sum co-array $S_{sum}=\left\{0,1,2,3,4,5\right\}$ in the end. $S_{u}=\left\{1,2,3,6\right\}$ can finally obtain a uniform liner difference co-array $S_{diff}=\left\{-5,-4,-3,-2,-1,0,1,2,3,4,5\right\}$. In this case, the consecutive DOFs of $S_{diff}$ is 11 (From −5 to 5).

### 3.2. Reformulation of Generalized Nested Array

A generalized nested array (GNA) consists of two sparse uniform linear subarrays [15]. One subarray consists of $M_{1}$ elements with element-spacing $\alpha d_{0}$ while the other subarray contains $M_{2}$ elements with element-spacing $\beta d_{0}$, where $d_{0}$ is the unit sensor spacing and will be omitted in the subsequent text. Here, α and β are flexible co-prime factors satisfying $1\leq \alpha \leq M_{2}$, $1\leq \beta \leq M_{1}+1$. The position set of GNA elements can be represented as
(10)$$\begin{array}{cc} S_{GNA}=\{1,1+\alpha , & \cdots ,1+\alpha M_{1},1+\alpha M_{1}+\beta ,\\ & \cdots ,1+\alpha M_{1}+\beta \left(M_{2}-1\right)\}. \end{array}$$

In [15], it has been demonstrated that the DOFs of GNA are related to the variables α, β, $M_{1}$ and $M_{2}$. Notably, the maximum DOFs of GNA are attained when $M_{1}=M_{2}$ or $M_{2}=M_{1}+1$ and one of the two factors α or β reaches its maximal value. Under this circumstance, the difference co-array of GNA is non-consecutive and results in a decrease in the mutual coupling leakage rate.

### 3.3. Design Principle for GNA-MIMO Radar

According to the GSDC concept [21], the virtual array elements of a MIMO radar can be computed by the operation sum co-array of the transmitter’s difference co-array and the receiver’s difference co-array. This approach allows us to utilize a GNA in the transmitter array with the maximum possible element spacing and an NA with an extension factor γ in the receiver array. Despite the non-continuous nature of their individual difference co-arrays, the sum co-array of these two difference co-arrays remains continuous. Consequently, this design strategy offers not only a large number of consecutive DOFs but also mitigates the mutual coupling effect.

The configuration of the proposed GNA-MIMO radar involves two steps:

(1) For the transmitter array, we employ an *M*-element GNA. The optimal values for variables are provided in Table 1 [15]. When *M* is even, we take the maximum value of α, β, that is $\alpha =M_{2}=M/2$, $\beta =M_{1}+1=M/2+1$. In this case, β=α+1, α and β satisfy the co-prime relation. However, if we take the maximum value of α, β when *M* is odd, α and β are not co-prime. So, we think $\alpha =M_{2}-1=\left(M-1\right)/2$. In this case, β=α+1, α and β also satisfy the co-prime relation. The difference co-array for the transmitter array, denoted as $D_{T}$, can be written as
(11)$$\begin{array}{cc} D_{T} & =D_{T}^{+}\cup D_{T}^{-}\\ D_{T}^{+} & =\{n-n^{\prime }|n\geq n^{\prime },n,n^{\prime }\in S_{T}\}\\ \; & =\{\alpha m_{1}+\beta m_{2}|0\leq m_{1}\leq M_{1},0\leq m_{2}\leq M_{2}-1\}. \end{array}$$
where $D_{T}^{+}$ and $D_{T}^{-}$ denote the non-negative and negative values in $D_{T}$, respectively.

(2) For the receiver array, we utilize an *N*-element NA with an extension factor $\gamma =M_{2}\left(M_{1}+1\right)$. If *N* is even, the number of elements for the two subarrays is $N_{1}=N_{2}=N/2$, respectively, otherwise, if *N* is odd, $N_{1}=\left(N-1\right)/2$ and $N_{2}=\left(N+1\right)/2$. The difference co-array for the receiver array, denoted as $D_{R}$, can be presented as [13]
(12)$$D_{R}=\gamma \left\{-\left(N_{2}\left(N_{1}+1\right)-1\right),\cdots ,0,1,\cdots ,N_{2}\left(N_{1}+1\right)-1\right\}.$$

In Figure 1, we present an instance of the proposed GNA-MIMO radar with a six-element transmitter array and a four-element receiver array. The position of the array element is determined based on the aforementioned design rules. Considering the symmetric property of difference co-array, only the non-negative part of the GSDC is plotted. Figure 1 illustrates that although elements in the transmitter and receiver arrays are sparsely located, the GSDC still has a long consecutive segment.

### 3.4. Properties of the GNA-MIMO Radar

**Proposition** **1.**
*The closed-form expression of attainable consecutive DOFs for the proposed GNA-MIMO radar is*

(13)
$$f=2\left(M_{2}\left(M_{1}+1\right)\right)\left(N_{2}\left(N_{1}+1\right)\right)-1.$$



**Proof.** The $D_{T}^{+}$ in (Equation 11) has the following properties:(a) There are $M_{2}\left(M_{1}+1\right)$ distinct integers in the set $D_{T}^{+}$.(b) $0\leq \alpha m_{1}+\beta m_{2}\leq \alpha M_{1}+\beta \left(M_{2}-1\right)$.(c) There are not such two values in $D_{T}^{+}$ that differ by a distance of $M_{2}\left(M_{1}+1\right)$. In other words, the inequality $\alpha m_{1}+\beta m_{2}-\left(\alpha m_{1}^{\prime }+\beta m_{2}^{\prime }\right)\neq M_{2}\left(M_{1}+1\right)$ holds true.Dividing both sides of the inequation by β, the expression $\frac{\alpha }{\beta }\left(m_{1}-m_{1}^{\prime }\right)\neq M_{2}+\left(m_{2}-m_{2}^{\prime }\right)$. We can see that $\frac{\alpha }{\beta }\left(m_{1}-m_{1}^{\prime }\right)$ must be a decimal number except for $m_{1}=m_{1}^{\prime }$ while the right side of the inequality is a positive integer. Therefore, the inequality definitely holds true.(d) The set $S^{\prime }=\left\{\mathrm{mod}\left(\alpha m_{1}+\beta m_{2},M_{2}\left(M_{1}+1\right)\right)|0\leq m_{1}\leq M_{1},0\leq m_{2}\leq M_{2}-1\right\}$ contains all integers from 0 to $M_{2}\left(M_{1}+1\right)-1$, where mod represents the modulo operation.From (a) (b) (c), we can conclude that the set $S^{\prime }$ also contains $M_{2}\left(M_{1}+1\right)$ distinct integers in the range $\left[0,M_{2}\left(M_{1}+1\right)-1\right]$. Therefore, the set $S^{\prime }$ contains all integers from 0 to $M_{2}\left(M_{1}+1\right)-1$.(e) The set $\left\{D_{T}^{+}-M_{2}\left(M_{1}+1\right)\right\}\cup D_{T}^{+}$ contains the set $S^{\prime }$ and is consecutive in the range $\left[0,M_{2}\left(M_{1}+1\right)-1\right]$.The GSDC for GNA-MIMO $S_{GSDC}$ has the following properties:(f) $S_{GSDC}=\left\{D_{T}^{+}\cup D_{T}^{-}\right\}+D_{R}=\left\{D_{T}^{-}+D_{R}\right\}\cup \left\{D_{T}^{+}+D_{R}\right\}$.(g) The set $D_{R}+D_{T}^{+}$ is consecutive in the range $\left[0,N_{2}\left(N_{1}+1\right)\gamma -1\right]$.The set $D_{R}+D_{T}^{+}$
$$\begin{array}{cc} \; & \subseteq \gamma \left\{-1,0,\cdots ,N_{2}\left(N_{1}+1\right)-1\right\}+D_{T}^{+}\\ & =\left\{-\gamma +D_{T}^{+}\right\}\cup D_{T}^{+}\cup \cdots \cup \left\{\gamma \left(N_{2}\left(N_{1}+1\right)-1\right)+D_{T}^{+}\right\}. \end{array}$$According to (e), the set $\left\{-\gamma +D_{T}^{+}\right\}\cup D_{T}^{+}$ contains [0,γ−1] and the set $D_{T}^{+}\cup \left\{\gamma +D_{T}^{+}\right\}$ contains [γ,2γ−1]. By inference, the above set certainly contains
$$\begin{array}{cc} \left[0,\;\gamma -1\right]\cup & \left[\gamma ,2\gamma -1\right]\cup \cdots \cup [\left(N_{2}\left(N_{1}+1\right)-1\right)\gamma ,\left(N_{2}\left(N_{1}+1\right)\right)\gamma -1]. \end{array}$$After merging, we can conclude that the set $D_{R}+D_{T}^{+}$ is consecutive in the range $\left[0,\left(N_{2}\left(N_{1}+1\right)\right)\gamma -1\right]$. Based on the property (f), the proposed GNA-MIMO radar ultimately obtains a virtual array from -*u* to *u*, where $u=(N_{2}\left(N_{1}+1\right))\gamma -1$. Consequently, the consecutive DOFs can be calculated as
(14)$$\begin{array}{c} f=2u+1=2M_{2}N_{2}\left(M_{1}+1\right)\left(N_{1}+1\right)-1 \end{array}$$□

### 3.5. Optimal *M* and *N*

Although the number of physical sensors in the transmitter and receiver arrays can be selected according to the specific requirements, the maximum number of consecutive DOFs can be realized by choosing optimal values of *M* and *N* given the number of total physical sensors *T*. Based on the arithmetic mean–geometric mean inequalities, the optimal values of *M* and *N* can be achieved by
(15)$$\left\{\begin{array}{c} M=N=T/2,\;\mathrm{if}\;T\;\mathrm{is}\;\mathrm{even}\\ M=(T-1)/2,N=(T+1)/2,\;\mathrm{if}\;T\;\mathrm{is}\;\mathrm{odd}. \end{array}\right.$$

## 4. Performance Analysis

In this part, we provide sufficient theoretical analysis to demonstrate the advantages of the proposed GNA-MIMO radar. First, four other MIMO radars including UCA-MIMO radar in [25], UGCA-MIMO radar in [28], NA-MIMO radar in [20] and GTNA-MIMO radar in [27] are adopted for comparison. Then, three aspects including the consecutive DOFs, the mutual coupling leakage and the Cramer–Rao Bound (CRB) are considered for analysis.

### 4.1. The Consecutive DOFs

Usually, the optimal array configuration and design methodology of the MIMO radar should be determined before comparing the consecutive DOFs of the different MIMO radars. Here, for the GTNA-MIMO radar, $\alpha =M_{1}M_{2}-M_{1}-M_{2}$ and β=α+1 and the optimal array configurations for the other types of MIMO radar can be found in the corresponding original text.

Table 2 lists the closed-form expression of consecutive DOFs for five types of MIMO radar. In the table, $M^{\prime }$ and $N^{\prime }$ represent two coprime numbers in the co-prime MIMO radar, where $M_{1}$ and $M_{2}$ represent the number of array elements in the transmit array of nested array MIMO radar and $N_{1}$ and $N_{2}$ represent the number of array sensors in the receive array of nested array MIMO radar. In general, MIMO radar with more DOFs will obtain a more accurate DOA result. Also, from the table, we can observe that the proposed GNA-MIMO radar has relatively high consecutive DOFs and achieves the same consecutive DOFs as the NA-MIMO radar in [20].

Figure 2 depicts the variation of attainable consecutive DOFs with a given number of physical sensors for these five different MIMO radars. It can be observed from Figure 2 that as the number of array elements increases, the consecutive DOFs continuously increase. Given the same number of physical sensors, the proposed GNA-MIMO radar can achieve the maximum consecutive DOFs as the NA-MIMO radar in [20] and exceeds other types of MIMO radar. Additionally, the solid lines in the figure represent that the nested array MIMO radars can take the number of array elements for all integers on the abscissa and the curves are smoother. However, the dashed lines in the figure represent that the co-prime MIMO radars can only take a few discrete integers on the abscissa, indicating that the design of nested array MIMO radar is more flexible.

### 4.2. The Mutual Coupling Leakage Rate

This subsection compares the coupling leakage rates of different MIMO radars under the optimal array element configuration. Generally, the smaller the mutual coupling leakage rate, the less influence the MIMO radar experiences in DOA estimation, resulting in more accurate DOA estimation results. The mutual coupling is built on (Equation 6) with $c_{0}=1$, $c_{1}=0.3e^{j\pi /3}$, B=100, $c_{l}=c_{1}e^{-j(l-1)\pi /8}/l$ for 2≤B≤100 [30].

Figure 3 compares the coupling leakage rates for these five MIMO radar systems when the total number of array elements is 12 and 24. From Figure 3, it can be observed that the proposed GNA-MIMO radar exhibits a lower coupling leakage rate compared to the other MIMO radars. Additionally, it can be observed that UCA-MIMO radar exhibits the highest mutual coupling leakage rate, mainly resulting in smaller element spacing and a significant mutual coupling effect. The mutual coupling leakage rate of NA-MIMO radar is also significant, especially when there are lots of array elements. The reason is that NA-MIMO radar contains a large number of adjacent array elements, which increases as the number of array elements increases.

### 4.3. The CRB Bound

The CRB bound means that the variance of an unbiased estimator can only approximate the CRB infinitely but not better than it. Thus, CRB provides a standard for the performance of an unbiased estimator. The CRB expression in this paper is derived according to the proof in the literature [22],
(16)$$\begin{array}{c} {\mathrm{CRB}}_{\theta }={\frac{1}{L}\left(F_{\theta }^{H}{⊓}_{F_{s}}^{⊥}F_{\theta }\right)}^{-1}, \end{array}$$
where $F_{\theta }={\left({R_{x}}^{T}⊗R_{x}\right)}^{-\frac{1}{2}}\hat{C}R_{s}$, ${⊓}_{F_{s}}^{⊥}=I-F_{s}{\left({F_{s}}^{H}F_{s}\right)}^{-1}{F_{s}}^{H}$, $F_{s}={\left({R_{x}}^{T}⊗R_{x}\right)}^{-\frac{1}{2}}A_{0}$, $\hat{C}={\hat{A}}^{*}⊙A+A^{*}⊙\hat{A}$ and $\hat{A}=\frac{\partial }{\partial \theta }A$.

Figure 4 shows the CRB curves of five MIMO radars where the SNR ranges from −15 dB to 15 dB and snapshots L=100. It can be observed from the figure that the CRB curve of the proposed GNA-MIMO radar has a smaller corresponding value than that of the other four MIMO radars. Therefore, the MIMO radar designed in this paper has the best DOA performance theoretically.

## 5. Simulation Results

In this part, we provide numerical simulation results to demonstrate the effectiveness and superiority of our proposed GNA-MIMO radar. First, we consider the optimal five MIMO radar configurations mentioned in the last section. Each of them is composed of 12 physical sensors. The mutual coupling is also built on (Equation 6) with $c_{0}=1$, $c_{1}=0.3e^{j\pi /3}$, B=100 and $c_{l}=0.3e^{-j(l-1)\pi /8}/l$ for 2≤B≤100 [30]. In the experiment, the signal power of the receiver array is fixed at 16 DBm. The transmitted signal is a Gaussian signal and the noise simulation of the channel is additive Gaussian white. Subsequently, the SS-MUSIC algorithm is utilized to estimate DOA. At last, the results of the MUSIC spectrum, the root mean square error (RMSE) curves and angle resolution performance are obtained.

### 5.1. MUSIC Spectrum

Figure 5 shows the spatial spectrum with the SS-MUSIC algorithm for four radar systems, except for the UCA-MIMO radar in [25]. Because of the limited consecutive DOFs of the UCA-MIMO radar, effective spatial spectra cannot be obtained. Assuming that there are K=31 incoherent targets in the far field evenly distributed from −30° to 30°. SNR = 3 dB; the number of snapshots L=100; the search range is from −31° to 31° and the step size is 0.01°. It can be seen from the figure that the MIMO radar designed in this paper can estimate incoherent targets far beyond the total number of physical array sensors. Moreover, the proposed GNA-MIMO radar exhibits more precise angular resolution and sharper spectral peaks.

### 5.2. The RMSE Performance versus SNR and Snapshots

We utilize the SS-MUSIC algorithm to research the different performances of DOA estimation with five MIMO radars. It is assumed there are K=6 incoherent targets impinging on the MIMO radar from angles [10°, 20°, 30°, 40°, 50°, 60°]. The RMSE of estimated DOAs can be calculated by [20]
(17)$$\begin{array}{c} \mathrm{RMSE}=\sqrt{\frac{1}{K\Gamma }{\left(\sum_{k=1}^{K}\sum_{\tau =1}^{\Gamma }\theta_{k}-{\hat{\theta }}_{k,\tau }\right)}^{2}}, \end{array}$$
where Γ is the number of Monte-Carlo trials, ${\hat{\theta }}_{k,\tau }$ denotes the τ-th trial of the *k*-th angle $\theta_{k}$ and $\theta_{k}$ represents the true DOA.

Figure 6 depicts the DOA estimation performance with different MIMO radars versus SNR, where the number of snapshots L=1000. Figure 7 illustrates the RMSE performance versus the number of snapshots, where SNR is set as 0 dB. It can be observed that the DOA estimation results become more and more accurate with the increase in snapshots and SNR. Specifically, under the condition of given snapshots and SNR, the proposed GNA-MIMO radar outperforms other MIMO radars due to its ability to provide pretty much more consecutive DOFs and a lower mutual coupling rate.

### 5.3. Resolution Performance

Assuming that there are two close uncorrelated targets located at $\theta_{1}$ and $\theta_{2}$ and the estimation results of these two angles are, respectively, denoted as ${\hat{\theta }}_{1}$ and ${\hat{\theta }}_{2}$. If $\theta_{1}$, $\theta_{2}$, ${\hat{\theta }}_{1}$, ${\hat{\theta }}_{2}$ satisfy that $|\theta_{1}-{\hat{\theta }}_{1}\left|<\right|\theta_{1}-\theta_{2}|/2$ and $|\theta_{2}-{\hat{\theta }}_{2}\left|<\right|\theta_{1}-\theta_{2}|/2$, we consider these two targets can be successfully resolved [21]. In this section, we assume $\Delta \theta =|\theta_{1}-\theta_{2}|=0.5^{\circ }$.

Figure 8 depicts the success rate of angle resolution with the variation of SNR when the number of snapshots is 10. Figure 9 demonstrates the success rate of angle resolution with the variation of snapshot number where SNR is −6 dB. From Figure 8 and Figure 9, it can be observed that as the SNR and snapshot number increase, the success rate of angle resolution improves. Obviously, under the same simulation conditions the proposed GNA-MIMO radar achieves the highest success rate of angle resolution.

## 6. Conclusions

This paper introduces a GNA-MIMO radar configuration with improved DOA performance. In the meantime, the closed-form expressions for the position of physical sensors and the attainable consecutive DOFs are derived. Simulation results verify the advantages of the proposed MIMO radar in DOA estimation performance and angle resolution using the SS-MUSIC algorithm. Moreover, our findings emphasize the importance of considering mutual coupling in MIMO radar design and highlight the efficacy of the proposed GNA-MIMO radar system in overcoming these challenges. This research opens up new possibilities for improving the performance of array signal processing techniques in practical applications. Furthermore, it is possible to substitute the receiver array with other sparse arrays to further increase DOFs while reducing the mutual coupling leakage rate.

## Figures and Tables

**Figure 1 sensors-24-03952-f001:**
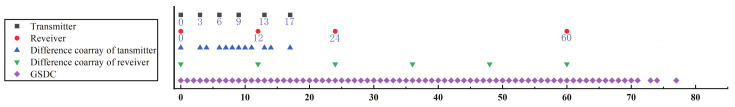
An example of a proposed radar array with a six-element transmitter array and a four-element receiver array. Only the non-negative part of the GSDC is given. The squares in the Transmitter and circles in Receiver are the actual positions of physical sensors while the others are virtual elements.

**Figure 2 sensors-24-03952-f002:**
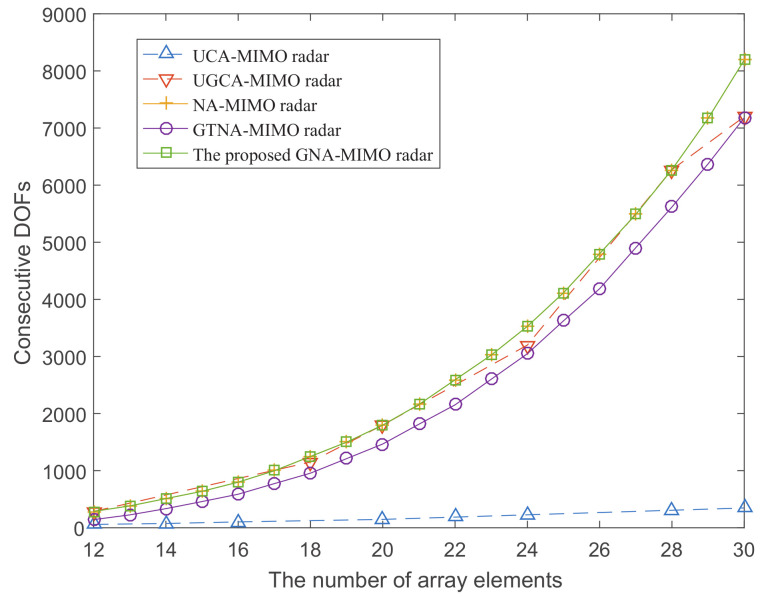
The attainable consecutive DOFs with given number of physical sensors.

**Figure 3 sensors-24-03952-f003:**
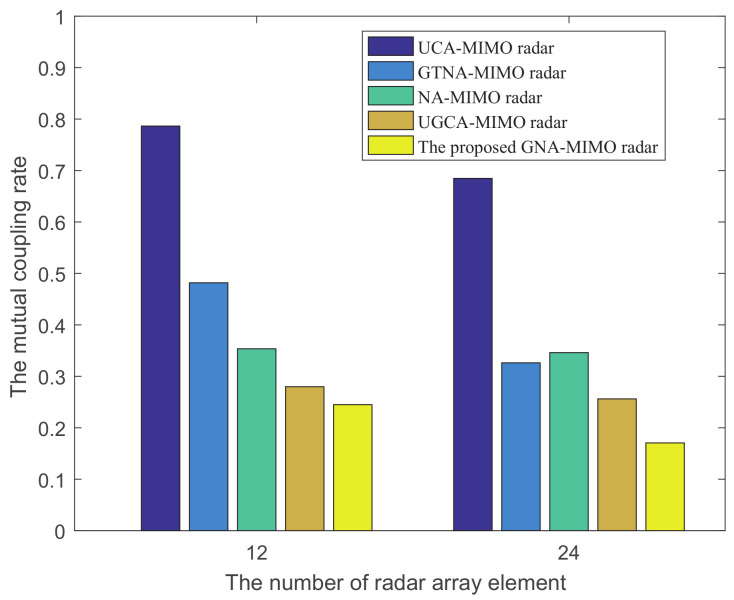
The coupling leakage rates for these five MIMO radar systems when the total number of array elements is 12 and 24.

**Figure 4 sensors-24-03952-f004:**
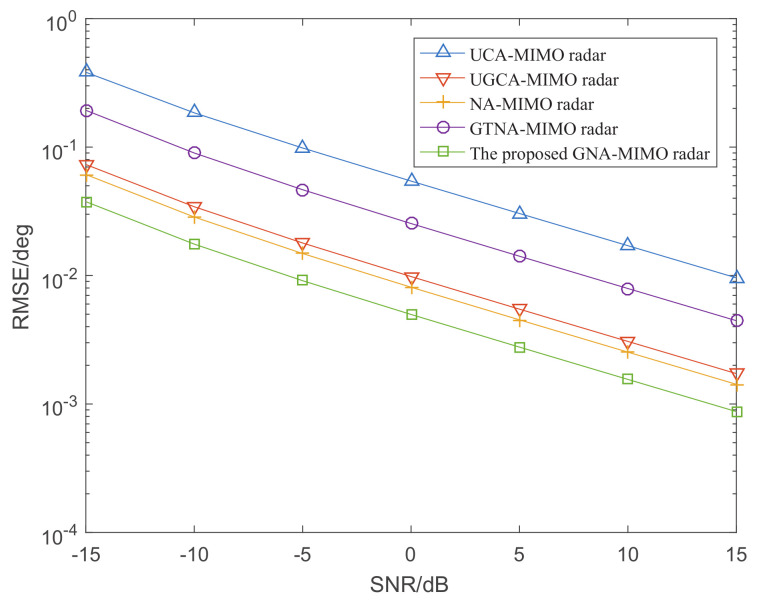
The CRB curves of five MIMO radars when the snapshots *L* = 100.

**Figure 5 sensors-24-03952-f005:**
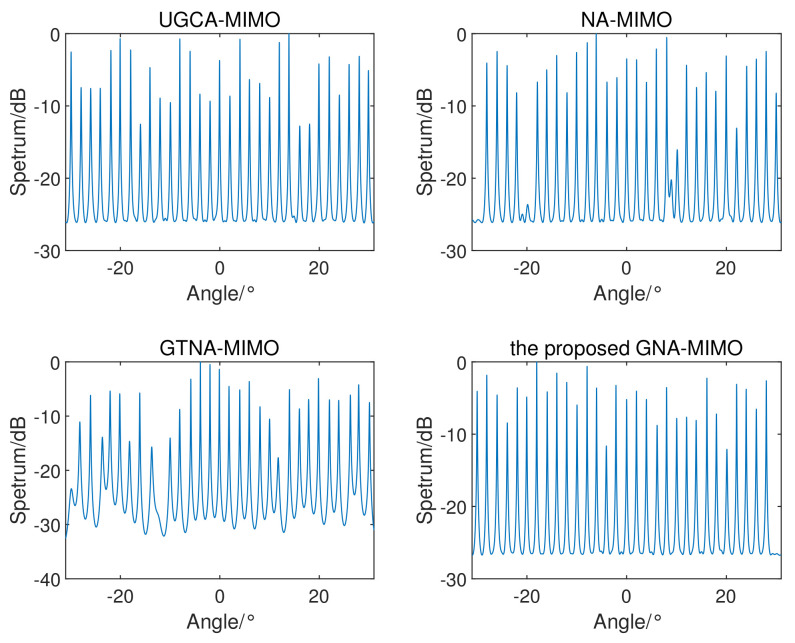
The spatial spectrum with SS-MUSIC when there are K=31 incoherent targets.

**Figure 6 sensors-24-03952-f006:**
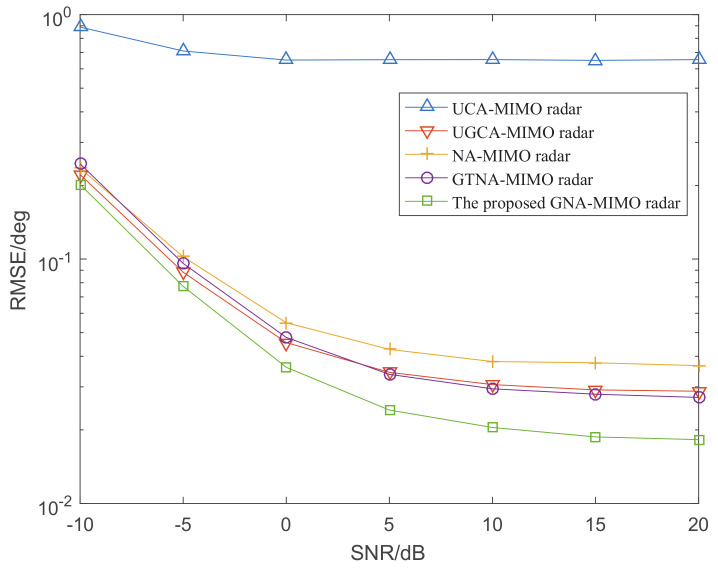
The RMSE performance versus SNR when L=1000.

**Figure 7 sensors-24-03952-f007:**
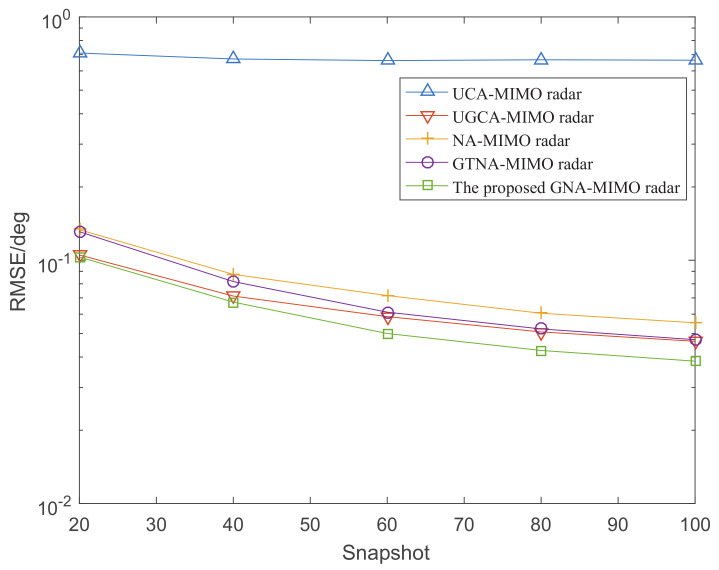
The RMSE performance versus the number of snapshots when SNR = 0.

**Figure 8 sensors-24-03952-f008:**
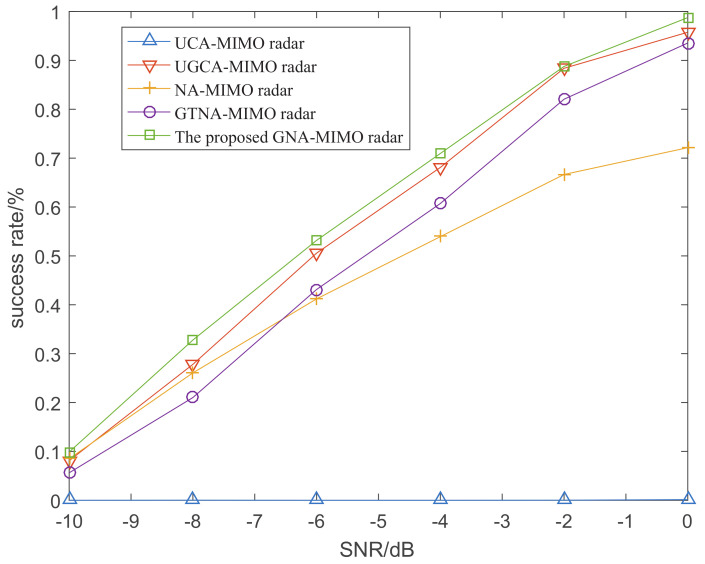
The rate of successful resolution versus SNR when L=10.

**Figure 9 sensors-24-03952-f009:**
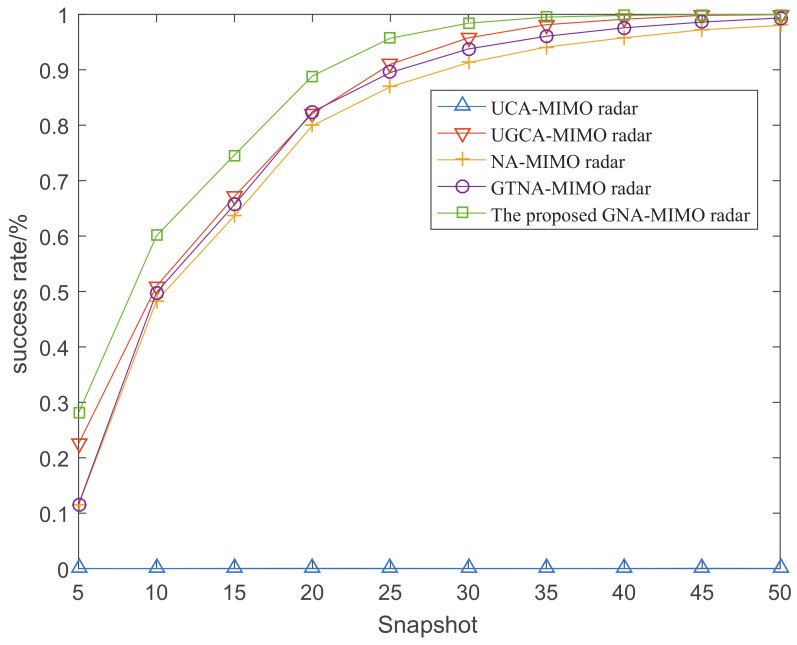
The rate of successful resolution versus the number of snapshots when SNR=−2.

**Table 1 sensors-24-03952-t001:** The Optimal Variables Values of $M_{1}$,$M_{2}$ and α,β.

When *M* Is	The Optimal $M_{1},M_{2}$	The Optimal α,β
Even	$M_{1}=M_{2}=M/2$	α=M/2, β=M/2+1
Odd	$M_{1}=\left(M-1\right)/2$, $M_{2}=\left(M+1\right)/2$	α=(M−1)/2, β=(M+1)/2

**Table 2 sensors-24-03952-t002:** The closed-form expression of attainable consecutive DOFs for five types of MIMO radar.

	The Total Number of Physical Radar Elements	The Closed-Form Expression of Consecutive DOFs
UCA-MIMO radar in [20]	$M^{\prime }+N^{\prime }$	2(3MN−M−N)−1
GTNA-MIMO radar in [19]	$M_{1}+M_{2}+N_{1}+N_{2}$	$2\alpha M_{2}\left(M_{1}-1\right)+2\beta N_{2}\left(N_{1}+1\right)$
NA-MIMO radar in [19]	$M_{1}+M_{2}+N_{1}+N_{2}$	$2M_{2}N_{2}\left(M_{1}+1\right)\left(N_{1}+1\right)-1$
UGCA-MIMO radar in [24]	$M^{\prime }+N^{\prime }$	$8{\left(MN\right)}^{2}-1$
The proposed GNA-MIMO radar	$M_{1}+M_{2}+N_{1}+N_{2}$	Equation (14)

$\alpha =M_{1}M_{2}-M_{1}-M_{2}.\beta =\alpha +1$. $M^{\prime }$ and $N^{\prime }$ are coprime.

## Data Availability

Data are contained within the article.

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
