# Peer review of "A Novel Generalized Nested Array MIMO Radar for DOA Estimation with Increased Degrees of Freedom and Low Mutual Coupling"

_sensors, 2024, doi:10.3390/s24123952_

Round 1
Reviewer 1 Report
Comments and Suggestions for Authors
The authors propose a MIMO radar architecture with low mutual coupling, evaluate its properties, and compare them with other existing architectures. While the paper may potentially contain a result worth publishing, the technical presentation needs to be significantly improved. Please consider the following recommendations:
1. Many key terms are mentioned, but never explained. Thus, the readers not working directly on MIMO radar would not be able to comprehend the ideas. For instance, DoF and consecutive DoF are mentioned a number of times, but it is never defined, nor explained why it is important for MIMO radar.
2. Having a picture like the one in Figure 1 would be very helpful to understand the paper. Are the transmitting and receiving antennas co-located? Does this create additional coupling?
3. Mathematical description is poor. Some vectors are capital letters, then later the same vector is small letter or vice versa. Kronecker product and tensor product use the same symbol?
4. l. 53 mentions that previous works ignored mutual coupling. This would be a great contribution, but I cannot see any results related to analyzing mutual coupling?
5. Claiming that Doppler shift can be neglected requires some justification., if the targets or the radar are moving. The reason why one can neglect multipaths is the narrow band assumption and free-space environment.
6. How to generated transmitted signal s(t)? In (6), is the index an integer i.e. the distance between antenna elements is an integer? How is (7) used anywhere to analyze coupling? Is A defined in (5) a matrix or vector?
7. It is unclear why vector form (9) of (8) is considered. It should be fully equivalent, so why to claim that (9) has more DoF than (8)?
8. Definitions 1, 2 and (10) - are these just sets? Do the authors imply that they consider a linear non-uniform array? The authors should show that the maximum DoF can be achieved and why for these settings, and how it affects mutual coupling.
9. Note that tables run away from the page, and in Figure 1, 6 vs. 4 elements does not corresponds to the text.
10. Ok, one can determine optimum M and N given T=M+N. But how to choose T?
11. The results in Table 1 and 2 should be derived or referenced.
12. Please describe Figure 5 how to understand the difference in the spectra.
13. How to understand the success rate in Section 5.3?
14. In Introduction when discussing the paper contributions, please distinguish between the proposed radar architecture, and its properties; the properties are no contribution, but e.g. derivation of properties can be considered to be contribution.
15. Other: why alpha and beta need to be co-primes? Radar does not provide closed form expressions. What is sensor redundancy?
Comments on the Quality of English LanguageSome sentences are not formulated following proper English grammar.
Reviewer 2 Report
Comments and Suggestions for Authors
Comments for Transmittal to Author:
The subject is of good theoretical significance and practical application value. The work is also clear from the mathematical point of view, and I think it to be free from basic errors and faulty expressions. The given theory is supported by numerical examples which demonstrate the accuracy and efficiency of the proposed method.
My comments to further improve the paper for publication are exhibited in the following:
1. This paper verifies that this method only uses Gaussian signal test for test cases that improve DOA accuracy. So can it be extended to other transmitted signal types and verified by other DOA estimation algorithms?
2. Please add the changes of array prevalence after MIMO radar virtual aperture expansion and differential array expansion. In addition, what is the result of the expression derivation of the array manifold matrix of the new virtual array?
3. Can this method be verified in the case of two-dimensional arrays?
Round 2
Reviewer 1 Report
Comments and Suggestions for Authors
The presentation has been somewhat improved, but it is still not sufficient.
1. Lines 23-28: saying that DoF is key parameter and it affects the performance explain why it is important, but does not explain what it is. Likewise, consecutive DoF remain undefined.
2. Kronecker product and tensor product still use the same symbol.
3. It is correct to say that if the mutual movements of radar and the target can be neglected, then Doppler shift can be neglected. However, in such a stationary scenario, radar would not be so useful as the distance does not change. The authors need to provide much more robust explanation why their analysis can ignore the Doppler shift.
4. The signal source s(t) remain undefined; would not this signal have strong effect on the radar performance? It can be certain pulse, or continuous etc.
5. In (6), how to choose the elements of the coupling matrix? How does the choice affect the performance? Why is it referred to as coupling rate, and not just coupling?
6. Line 19: d_n is position of antenna, scalar value in 3D space?
7. Where is (7) being used in the analysis? I is identity matrix, not unit matrix.
8. l. 124: how is the matrix S(t) defined?
9. l. 128-130: where the virtual array elements come from, i.e how are they defined? How to understand single snapshot? Were there multiple snapshots assumed before?
10. Definitions in Section 3.1 rely on the array S_a and S_b, but these were never defined. Likewise, N_1 and N_2 are not defined.
11. l. 181: whether the antennas are closely located is a matter of personal opinion unless some precise definition is provided.
12. Figure 1 must be much better explained to understand what is given, what is representing physical array, what may be equivalent model with virtual elements.
13. Why part of the mathematical expressions is in the text, and the rest displayed in Section 3.4? Why the expression starts with a union symbol? Why in (14) the product is denoted by '*', but not elsewhere?
14. It is unclear how Figure 3 was generated.
15. Description of Figure 5 is insufficient (why the proposed radar architecture is better?).
16. If the transmitter and receiver antennas are co-located, how to deal with their mutual coupling? I can imagine it can be rather strong.
In general, please note that many explanations to the reviewers should be also added to the paper for other readers.
Comments on the Quality of English Language
There are many sentences that are grammatically incorrect, e.g. missing object, or incomplete. The sentence should never start with 'And ..'. I can imagine that a professional proofreading may be required.
